# Mega-Dose Vitamin C Ameliorates Nonalcoholic Fatty Liver Disease in a Mouse Fast-Food Diet Model

**DOI:** 10.3390/nu14112195

**Published:** 2022-05-25

**Authors:** Seoung-Woo Lee, Young-Jin Lee, Su-Min Baek, Kyung-Ku Kang, Tae-Un Kim, Jae-Hyuk Yim, Hee-Yeon Kim, Se-Hyeon Han, Seong-Kyoon Choi, Sang-Joon Park, Tae-Hwan Kim, Jin-Kyu Park

**Affiliations:** 1Department of Veterinary Pathology, College of Veterinary Medicine, Kyungpook National University, Daegu 41566, Korea; pyrk2000@gmail.com (S.-W.L.); bnm3448123@naver.com (Y.-J.L.); suminbaek@naver.com (S.-M.B.); tukim92@naver.com (T.-U.K.); dlajh7676@gmail.com (J.-H.Y.); khyon1206@dgist.ac.kr (H.-Y.K.); thkim56@knu.ac.kr (T.-H.K.); 2Laboratory Animal Center, Daegu-Gyeongbuk Medical Innovation Foundation, Daegu 41061, Korea; qzpmqzpm@naver.com; 3Core Protein Resources Center, Daegu-Gyeongbuk Institute of Science and Technology (DGIST), Daegu 42988, Korea; cskbest@dgist.ac.kr; 4Department of Media and Communication, Hanyang University, Seoul 04763, Korea; han382@daum.net; 5Laboratory of Veterinary Histology, College of Veterinary Medicine, Kyungpook National University, Daegu 41566, Korea; psj26@knu.ac.kr

**Keywords:** vitamin C, nonalcoholic fatty liver disease (NAFLD), uric acid, kidney, fructose

## Abstract

In previous studies, the increasing clinical importance of nonalcoholic fatty liver disease (NAFLD) has been recognized. However, the specific therapeutic strategies or drugs have not been discovered. Vitamin C is a water-soluble antioxidant and is a cofactor in many important biosynthesis pathways. Recently, many researchers have reported that the mega-dose vitamin C treatment had positive effects on various diseases. However, the precise relationship between mega-dose vitamin C and NAFLD has not been completely elucidated. This study has been designed to discover the effects of mega-dose vitamin C on the progression of NAFLD. Twelve-week-old wild-type C57BL6 mice were fed chow diets and high-fat and high-fructose diet (fast-food diet) ad libitum for 11 weeks with or without of vitamin C treatment. Vitamin C was administered in the drinking water (1.5 g/L). In this study, 11 weeks of the mega-dose vitamin C treatment significantly suppressed the development of nonalcoholic steatohepatitis (NASH) independently of the catabolic process. Vitamin C supplements in fast-food diet fed mice significantly decreased diet ingestion and increased water intake. Histopathological analysis revealed that the mice fed a fast-food diet with vitamin C water had a mild renal injury suggesting osmotic nephrosis due to fructose-mediated purine derivatives. These data suggest that the mega-dose vitamin C treatment suppresses high-fructose-diet-mediated NAFLD progression by decreasing diet ingestion and increasing water intake.

## 1. Introduction

Nonalcoholic fatty liver disease (NAFLD) is a chronic metabolic disorder and frequently coexists with obesity, diabetes, and insulin resistance. It is generally defined as the accumulation of hepatic triglycerides in excess of 5% of the liver weight without alcoholic intake, viral infection, and drug treatments [1,2]. NAFLD is a disease which has a broad spectrum, ranging from mild forms of NALD including nonalcoholic fatty liver (NAFL, steatosis) and nonalcoholic steatohepatitis (NASH), to irreversible end-stage liver diseases such as cirrhosis and hepatocellular carcinoma [1]. In previous research, a sedentary lifestyle and Western diet characterized by a high content of refined grains, sugar, and saturated fat, are spreading in both developing and developed countries, and the prevalence of chronic metabolic disorders has increased [2]. As a result, NAFLD has become one of the most common chronic liver diseases that has a 25.24% incidence rate worldwide [3]. Because the global burden of NAFLD is constantly increasing, there has been much research about the specific drugs and therapeutic strategies to treat NAFLD. However, the precise pathogenesis and specific therapeutic strategies of NAFLD have not been fully understood yet, and additional research is urgently needed.

Vitamin C is a water-soluble vital nutrient and is essential for the growth, development, and tissue repair of animals through collagen synthesis [4]. Vitamin C is also a coenzyme with a crucial role in many biosynthesis pathways and immune functions [5]. Vitamin C is a strong antioxidant and reduces oxidative-stress-induced cellular damage by inhibition of superoxide radicals [6]. Vitamin C is non-toxic, and the long-term safety of vitamin C intake has already been reported in many investigations [7].

According to previous studies, mega-dose vitamin C usage, characterized by doses over the recommended daily amount, is considered a potential therapy for many diseases, and many researchers have reported that the mega-dose vitamin C treatment demonstrates a high therapeutic potential for various diseases. In previous animal model study, mega-dose vitamin C treatment (Intravenous, 150 g per 40 kg over 7 h) drastically attenuated multi-organ dysfunction in a sheep model of sepsis caused by gram-negative bacteria [8]. In human beings, the mega-dose vitamin C treatment (Intravenous, 60 g) also exhibited potential therapeutic effects on COVID-19 patients [8]. Other researchers have also suggested that mega-dose vitamin C (Intravenous, 5–45 g per day and/or Oral, 5–20 g per day) might inhibit growth of tumor cells by enhancing human immune responses [9]. Other researchers also reported that the negative correlation between mega-dose vitamin C treatments and attenuated uric acid related diseases in human beings [10,11]. Moreover, any adverse effects of the mega-dose vitamin C treatment were not reported in any of these studies [7,8,9,10,11]. Given that, it is hypothesized that mega-dose treatment with vitamin C might have beneficial effects on metabolic disorders. However, the precise mechanisms between the mega-dose vitamin C treatment and NAFLD pathogenesis have not been completely studied yet. Therefore, we planned animal experiments to study the effects of mega-dose vitamin C on NAFLD.

## 2. Materials and Methods

### 2.1. Animal Experimental Design

Twelve-week-old male C57BL/6N mice (*n* = 20) were housed at room temperature (22 ± 3 °C) with a relative humidity of 50% ± 10%, a 12 h light-dark cycle (lights on at 8:00 AM), and free access to food and water. The mice were divided into four groups (*n* = 5) and fed the study diets for 11 weeks: a chow diet group (Chow), chow diet and vitamin C treated group (Chow + VC), a fast-food diet group fed a Western diet with sugar water (FFD), and a fast-food diet with vitamin C treated group (FFD + VC). The mice in the chow groups were used as controls and received a chow diet (Safe diet, SAFE D40, Rosenberg, Germany). The FFD mice groups received a Western diet (Research diet, D12079B, New Brunswick, NJ, USA, 41 kcal% from fat and 43 kcal% from carbohydrate) and sugar water solution (23.1 g/L d-fructose + 18.9 g/L d-glucose in filtered tap water). Additionally, vitamin C (1.5 g/L in filtered tap water) was provided to the chow + VC and FFD + VC mice groups for the entire study period. The food and water intake were checked daily, and body weight was measured three times per week. The ratio of increased body weight was calculated using the following equation: (final day body weight—initial body weight/initial body weight) X 100). Additionally, Twelve-week-old male C57BL/6N mice (*n* = 5) were housed at the same condition as described above. These mice were fed by a chow diet and vitamin C contained in filtered tap water for 70 weeks to confirm safety of long-term mega-dose vitamin C treatment. Animal use and procedures were approved by the Kyungpook National University Institutional Animal Care and Use Committee (IACUC, approval number 2017-0112, 2021-0175 and 2021-0176).

### 2.2. Histopathology

The mouse tissue samples were collected at 11 and 70 weeks and fixed in a 10% neutral buffered formalin. After fixation, the tissue samples were processed per routine and embedded in paraffin wax. The paraffin blocks were cut into 3 to 5 μm thick sections for hematoxylin and eosin (H&E) staining. A histological evaluation of NAFLD was performed using the NASH Clinical Research Network (CRN) grading system [12]. Renal tubular injury was assessed as a proportion of damaged renal tubules (Table 1) [13]. The average adipocyte size was determined based on five random fields at 200 times magnification for each slide.

### 2.3. Primary Antibodies and Chemicals

The following antibodies were used: anti-β-actin (Santa Cruz, sc-47778, Santa Cruz, CA, USA; Sigma, A1978,St. Louis, MO, USA), anti-phspho AMP-activated protein kinase (p-AMPK) (Cell Signaling Technology, 2535S, Danvers, MA, USA), anti-AMPK (Cell Signaling Technology, 2532S, Danvers, MA, USA), anti-PPAR-α (Santa Cruz, sc-398394, San Francisco, CA, USA), and anti-myeloperoxidase (Invitrogen, PA5-16672, Invitrogen, Waltham, MA, USA). The following chemicals were used: vitamin C (DUKSAN, 832, Ansan, Korea), fructose (SANCHUN, F0366, Yeosu, Korea), and glucose (JUNSEI, 64220S0601, Tokyo, Japan).

### 2.4. Immunohistochemistry

For the immunohistochemistry analysis, paraffin-embedded slides were deparaffinized in toluene and rehydrated in graded alcohol series (70, 80, 90, 95, and 100%) and distilled water. After rehydration, antigen retrieval was performed in a solution of 3% hydrogen peroxide in methanol solution at room temperature (25 °C) for 35 min and steamed for 30 min in 10 mmol/L citrate buffer for heat-induced antigen retrieval. After retrieval, the slides were cooled at room temperature for 2 h and were incubated first with a blocking solution (Life Technologies, Frederick, MD, USA) at room temperature for 1 h, then the sections were incubated with the primary antibody overnight at 4 °C. After incubation, the sections were washed in phosphate buffered saline (PBS), and the sections were incubated with a broad-spectrum secondary antibody and streptavidin-horseradish peroxidase conjugate (Life Technologies, Frederick, MD, USA) at room temperature for 10 min. The antigen-antibody complex was visualized with a diaminobenzidine (DAB) (Vector Laboratories, Burlingame, CA, USA). The sections were counterstained with 10% hematoxylin for 2 min and were dehydrated with an alcohol series and toluene.

### 2.5. Western Blot Analysis

The frozen liver tissue samples were homogenized in a lysis buffer containing 0.1 mM sodium vanadate, a protease inhibitor cocktail tablet (Roche, Mannheim, Germany), pefabloc SC (Roche, Mannheim, Germany), sodium fluoride, and sodium pyrophosphate. The whole liver lysates were measured using a DC Protein Assay Kit (Bio Rad, Hercules, CA, USA). Equal amounts of protein (50 μg) were loaded and separated by 10% of sodium dodecyl sulfate-polyacrylamide gel electrophoresis (SDS-PAGE). Subsequently, the protein samples were transferred to polyvinylidene difluoride (PVDF) membranes (Millipore, IPVH00010, Billerica, MA, USA). The membranes were blocked in 5% skim milk for 1 h and incubated with the primary antibodies overnight at 4 °C. After washing with Tris buffered saline containing 0.1% Tween-20, membranes were incubated with a horseradish-peroxidase-conjugated goat-anti-rabbit (Calbiochem, 401393, San Diego, CA, USA) or goat-anti-mouse (Calbiochem, 401253, San Diego, CA, USA) antibody for 1 h at room temperature. The membranes were visualized using the ProNA™ ECL Ottimo (Translab, Seoul, Korea), and the images were acquired using the Amersham™ Imager 680 (GE Healthcare, Bjorkgatan, Sweden). β-actin was used as a loading control.

### 2.6. Quantitative RT-PCR

Liver RNAs were obtained using TRIzol Reagent (Invitrogen, Carlsbad, CA, USA), and the RNA concentrations were measured with a ND-1000 spectrophotometer (NanoDrop Technologies, Wilmington, DE). The extracted RNAs were used as a templet for cDNA synthesis using the RT Prime Kit (ELPIS Biotech, EBT-1520, Daejeon, Korea) with M-MLV Reverse Transcriptase, random hexamers, and oligo dT. Synthesized cDNAs were mixed with the TOPreal™ qPCR 2X PreMIX (Enzynomics, RT500M, Daejeon, Korea) with 5 pmol of primers. 18S ribosomal RNA was used as an internal control to normalize mRNA expression. The following mouse primers were used: 18s, IL-1β, IL-6, CD-11b, and ICAM-1. The sequence and source of the primers used in the present study are listed in Table 2 [14,15,16,17,18].

### 2.7. Serum Biochemistry for Measuring Triglyceride, Free Fatty Acid, ALT, and Vitamin C Levels

Blood was obtained from the caudal vena cava and collected into an eppendorf tube for 30 min at room temperature. Serum samples were centrifuged at 3000 rpm for 15 min at 4 °C and stored at −80 °C. The serum levels of alanine transaminase (ALT), aspartate aminotransferase (AST), alkaline phosphatase (ALP), glucose, and free fatty acid (FFA) were measured with an automated analyzer (Toshiba corporation, TBA-120FR, Tokyo, Japan).

### 2.8. Extraction and Measurements of Hepatic Triglycerides and Free Fatty Acids

Snap-frozen liver tissues were placed in isopropanol solution (Sigma, 278475, St. Louis, MO, USA) and incubated overnight at 4 °C. The samples then were centrifuged at 10,000 rpm for 15 min at 4 °C to aspirate the supernatants. The collected supernatants were measured with a Triglyceride L-type Kit (Wako, Osaka, Japan) and Free Fatty Acid Assay Kit (Biomax, Seoul, Korea) in accordance with the manufacturers’ instructions.

### 2.9. Statistical Analysis

All data obtained from the experiments were expressed as the mean ± standard deviation. The D’Agostino-Pearson normality test was used to confirm the normal distribution of data. Unpaired Student’s t-test, Mann-Whitney U test or Kruskal-Wallis one-way analysis of variance (Anova) on ranks were used to examine the variables that did not show normal data distribution. Body weight, caloric intake, water intake and diet ingestion were analyzed by two-way repeated ANOVA with the Bonferroni post-tests (time and treatment as the variables). All the statistical analysis were determined by GraphPad prism version 5 (GraphPad Software Inc., San Diego, CA, USA) and GraphPad InStat statistical package, version 3, (GraphPad Software Inc., San Diego, CA, USA).

## 3. Results

### 3.1. Mega-Dose Vitamin C Treatment Significantly Decreased Fast-Food-Diet Mediated Body Weight Gain

The mice were fed a chow diet or a fast-food diet with or without vitamin C for 11 weeks. (Figure 1A) The FFD + VC group exhibited lower body weight gain compared with the FFD group for the experimental period (Figure 1B). The FFD + VC group exhibited significantly decreased body weight (*p* value = 0.0015) (Figure 1C). Moreover, the FFD + VC groups demonstrate an almost equal increased body weight ratio (*p* value against to FFD group = 0.0045) with those of the chow-diet fed mice groups (Figure 1D). These data indicated that the mega-dose vitamin C treatment significantly suppressed the FFD-mediated body weight gain in mice. Microscopically, the FFD + VC group demonstrated significantly decreased average adipocyte size compared to the FFD group (Figure 1E,F). However, the caloric intake per body weight was equal in all the FFD-fed groups for the experimental period (Figure 1G). These results suggest that the mega-dose vitamin C treatment markedly decreased body weight gain in the FFD mice groups.

### 3.2. Mega-Dose Vitamin C Treatments Reduced Hepatic Triglyceride Accumulation in the Fast-Food-Diet Fed Mice

In the gross examination, the livers of the Chow and Chow + VC groups exhibited a shiny and reddish surface color, while the livers of FFD group showed a larger size, pale surface color, and multifocal white spots suggesting severe fat accumulation in the liver. Interestingly, the livers from FFD + VC group exhibited smaller size, reddish surface color and smoother appearance compared with those in the FFD group despite the FFD-diet ingestion (Figure 2A). Additionally, the FFD + VC mouse group showed decreased liver weights compared to the FFD mouse group (Figure 2B,C). In contrast, both Chow and Chow + VC groups exhibited similar gross appearance and liver weights (Figure 2A–C). In biochemical assays, the hepatic FFA, which is a main precursor for liver triglyceride synthesis, levels were significantly decreased in the FFD + VC group compared to the FFD group (Figure 2C). The hepatic triglyceride levels were significantly increased in all FFD-fed mice groups compared to chow-diet fed groups suggesting FFD-mediated NAFLD progression. However, the FFD + VC group exhibited notably decreased liver triglyceride levels compared to FFD group (Figure 2D). These data indicated that the vitamin C treatment selectively attenuated fast-food-diet mediated fat accumulation in liver.

### 3.3. Mega-Dose Vitamin C Treatment Attenuated NAFLD Progression Microscopically

Light microscopic examination revealed that the FFD-fed mice had severe steatosis lesions compared to the chow-diet-fed mice groups (Figure 3A). However, the FFD + VC group exhibited significantly ameliorated steatosis lesions despite of 11 weeks of FFD feeding. Additionally, the FFD + VC group exhibited notably decreased Oil-Red-O positive areas compared to the FFD group (Figure 3B). These data suggested that vitamin C treatment significantly decreased FFD-mediated liver fat accumulation in the mice. We also assessed the NAFLD progression of each mouse group using the NASH-CRN grading system. The histopathological examinations of the liver tissue samples revealed that after 11 weeks of FFD feeding the mice had developed NASH, a relatively late stage of NAFLD, while the FFD + VC mice group exhibited an ameliorated NAFLD progression that was similar to the chow-diet-fed mice groups (Figure 3C–F). These data indicated that the vitamin C treatment significantly decreased not only FFD-induced triglyceride accumulation in hepatocytes but also NAFLD progression.

### 3.4. Mega-Dose Vitamin C Treatment Attenuated Liver Injuries in the FFD-Fed Mice

In the biochemical analysis, the FFD-fed group exhibited notably increased serum liver enzyme levels such as ALT, ALP, and AST compared to the chow-diet-fed groups. However, the FFD + VC group exhibited notably decreased serum ALT, ALP, and AST levels despite the fast-food diet feeding (Figure 4A–C). These data demonstrate that vitamin C attenuated FFD-mediated liver injuries. However, serum levels of nutrients including glucose and FFA were similar in all FFD-fed mice groups despite the serum liver enzyme differences between the FFD and FFD + VC groups (Figure 4D,E). We are confident that nutrient absorption was not affected by vitamin C supplementation in the FFD-fed groups. Vitamin C treatment showed an improvement in the serum liver enzyme levels as well as attenuated NASH lesions in FFD-fed mice.

### 3.5. Mega-Dose Vitamin C Treatment Protects the Liver from NASH-Mediated Inflammatory Responses

Next, we performed immunohistochemistry using anti-MPO antibodies, which are a major marker of neutrophils. In immunohistochemistry staining, the FFD group showed many neutrophils, while the FFD + VC group demonstrated significantly decreased infiltration of inflammatory cells (Figure 5A,B). These data indicate that vitamin C treatment reduced infiltration of inflammatory cells into the liver parenchyma. Moreover, the FFD + VC group also exhibited significantly decreased inflammatory gene expression compared than FFD group (Figure 5C,F). This suggests that vitamin C treatment ameliorates the NASH-mediated inflammatory response.

### 3.6. Mega-Dose Vitamin C Treatment Attenuated NASH Progression Independent of Catabolic Pathways

Because the FFD + VC group showed significantly ameliorated NASH progression compared to the FFD group, we hypothesized that the mega-dose vitamin C treatment accelerates the catabolic process. However, the immunoblot analysis revealed that the FFD + VC group exhibited almost equal AMPK phosphorylation levels compared to the FFD group (Figure 6A,B). Since phosphorylated AMPK is a crucial factor in energy metabolism including catabolism and ATP synthesis, these data suggest that vitamin C mega-dose treatments in FFD mouse groups was not closely related to the catabolic process. Immunoblot analysis also revealed that there were no significant differences in PPAR-α expression between FFD + VC and FFD groups (Figure 6C,D). These data indicate that the mega-dose vitamin C treatment attenuated NASH progression independently of the catabolic process in the liver.

### 3.7. Mega-Dose Vitamin C Treatment Inhibits Diet Ingestion via Increased Water Intake with Compensatory Increases in Urine Volume

In this study, the mega-dose vitamin C treatment significantly ameliorated the NASH progression caused by a prolonged fast-food diet. The immunoblot analysis demonstrated that catabolic activity was not changed by mega-dose vitamin C supplementation. Notably in this study, the FFD + VC group demonstrated drastically increased water intake compared to the FFD group, while diet intake was significantly decreased by vitamin C treatment in the FFD-fed mouse group. (Figure 7A,B). The FFD + VC group demonstrated increased urine volume compared to the FFD group for experimental period (data not shown). The following histopathologic assessment revealed mild renal tubular injuries in FFD + VC group compared to FFD group, suggesting increased osmotic pressures in FFD + VC mice than FFD group (Figure 7C,D). These data insisted that the mega-dose vitamin C treatment significantly attenuated FFD-mediated NAFLD progression via inhibiting diet intake and increasing water intake in mice.

## 4. Discussion

In the present study, we evaluated the therapeutic effects of mega-dose vitamin C application in diet induced NAFLD and obesity animal models. For the entire experimental periods, the FFD + VC group exhibited significantly lower body weight compared to the FFD group. (Figure 1), and the hepatic fat contents including triglyceride and FFA were also significantly reduced by vitamin C supplements (Figure 2). The following histopathologic examination revealed that the FFD + VC group had drastically attenuated NASH lesions compared to the FFD group. The total NASH-CRN grade scores were significantly decreased by vitamin C treatments in the FFD-fed mice groups (Figure 3). Serum biochemistry levels demonstrated that the hepatic enzymes ALT, ALP, and AST were elevated by 11-weeks of FFD feeding, while the serum levels of glucose and FFA were not affected by vitamin C treatments (Figure 4). These data indicated that the therapeutic effects of mega-dose vitamin C are independent of nutrient absorption. In previous studies, prolonged high-carbohydrate and high-fat diets induce NASH development not only triglyceride accumulation in hepatocytes but also through infiltration of inflammatory cells into the liver parenchyma. [19] Consequently, Immunohistochemistry and qPCR results demonstrated that the mega-dose vitamin C treatment protects the liver from not only triglyceride accumulation but also infiltration of inflammatory cells and inflammatory cytokines (Figure 5). Immunoblot analysis demonstrated that the mega-dose vitamin C application did not increase the level of expression of proteins involved in catabolic activities in the FFD-fed mice (Figure 6). Therefore, we pose the question: how did the mega-dose vitamin C treatment significantly attenuate NAFLD progression independently of energy degradation? Interestingly, in this study, the FFD + VC group showed significantly higher water intake and lower food intake than FFD group (Figure 7A,B). Interestingly, the FFD + VC group was observed with notably increased urine volume (data not shown) compared to the FFD group. According to previous study, excessive intake of fructose-rich diet increased the serum levels of purine derivatives including uric acid, allantoin and urea, and these purine derivatives are mainly excreted by the renal system in rodents [20,21]. Urea is a water-soluble end product of purine degradation. Since these solutes can cause an osmotic diuresis and osmotic nephrosis, we hypothesized that the FFD + VC group had increased excretion of purine derivatives including uric acid, allantoin, and urea in the urine compared to those in the FFD group [22]. This hypothesis is consistent with the microscopic findings in the FFD + VC mice with prominent cytoplasmic vacuoles and swelling in cells of the proximal renal tubule suggesting mild osmotic nephrosis (Figure 7C,D) [23]. Subsequently, the water intake, diet ingestion and histopathologic lesions were not affected by the mega-dose vitamin C treatment in the chow-diet fed mice groups (Chow and Chow + VC) (Figure 7A–D). These results indicate that the mega-dose vitamin C treatment only affects the mice which were fed a fructose-rich diet for 11 weeks. Taken together, we can confirm that the mega-dose intake of vitamin C significantly increased the water intake of the FFD-fed mice with an expected increase in urine volume due to enhanced removal of FFD-mediated purine derivatives, and this increased water intake in the FFD + VC group ultimately inhibited NASH progression via reduced diet intake compared to FFD group.

Most of mammals, except for primates, excrete excessive purine derivatives to allantoin and urea which are more water-soluble solutes than uric acid via enzymatic reactions by uricase [24]. Therefore, it is assumed that the mega-dose vitamin C treatments enhance uricase activity in FFD-fed mice. Interestingly, there have been many studies using uricase or uricase recombinant to treat uric acid related disease in human beings [25,26]. The implications this previous research and the data in the present study suggest that a combination of uricase treatment and the mega-dose vitamin C treatment might have a synergistic effect in treating uric acid related disease in human beings. 

In the present study, the FFD + VC mice exhibited mild kidney injuries compared to the other mice groups (Figure 7). Additionally, there were few studies reported that the mega-dose vitamin C might have adverse effects [27]. Therefore, at first, we performed animal experiments to confirm the safety of the mega-dose vitamin C treatment in animals. As a result, the wild-type C57BL/6 mice treated by vitamin C at the same dose used in the present study for 70 weeks did not have any significant liver and kidney damage (Appendix A). Consequently, we can confirm the safety of the mega-dose vitamin C treatment.

Although there were many published studies regarding the therapeutic benefits of vitamin C in various diseases, this study focused on the previously unreported relationship between mega-dose vitamin C and NALFD. Generally, the water-soluble nutrients like vitamin C are absorbed from the small intestine and directly taken into the liver via the portal vein. Therefore, we hypothesized that the mega-dose vitamin C treatment might be efficacious more in the liver than in other organs. Notably, in this study, mega-dose vitamin C attenuated the FFD-mediated NAFLD. Since the FFD animal model has a similar pathogenesis to human NAFLD, it may be reasonable to assume that vitamin C may have a therapeutic effect in human NAFLD [28].

In the present study, we thoroughly investigated the effects of the mega-dose vitamin C treatment in NAFLD progression. However, there were some limitations of our current experiments. First, we suggested that the mega-dose vitamin C treatment accelerated degradations of the FFD-mediated purine derivatives in mice. However, the precise mechanisms of the mega-dose vitamin C treatment in uricase activities are still unknown. Seconds, all results observed in the present study come from animal experiments, and thus it can not be directly applied to human beings because of the species differences between human beings and mice. Given that, it seems that further studies are needed to confirm and expand these findings.

## 5. Conclusions

In summary, we have demonstrated that the mega-dose treatment of vitamin C significantly attenuated FFD-mediated NAFLD by decreasing diet ingestion due to increasing water intake. These data also suggested that the mega-dose vitamin C treatment facilitated the excretion of FFD-mediated purine derivatives in mice. Taken together, our study showed the possible beneficial effects of the mega-dose vitamin C treatment in NAFLD, and we anticipate that the results from the present study would lead to a better understanding of the relationship between the mega-dose vitamin C treatment and NAFLD.

## Figures and Tables

**Figure 1 nutrients-14-02195-f001:**
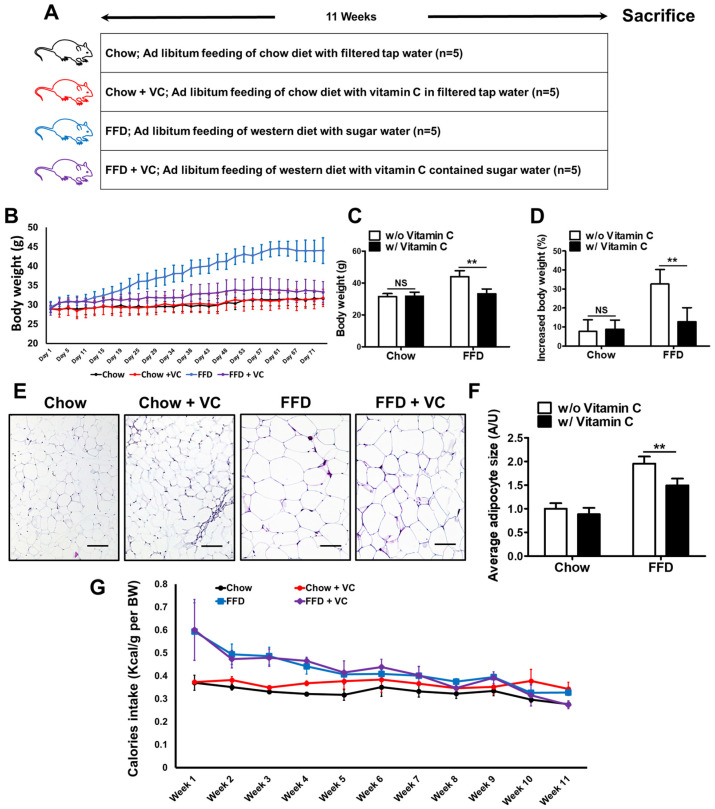
Comparison of body weight and calories intake between each mouse group. (**A**) Schematic diagram of the experimental protocol used to assess the effect of mega-dose vitamin C treatment on NAFLD (**B**) The body weight data of whole mice groups for entire experimental periods. (**C**) The body weight of final day; and (**D**) The increased body weight ratio (%) (**E**) Representative images of hematoxylin and eosin (H&E) staining; and (**F**) relative size of adipocytes. (**G**) The calories intake per body weight levels in whole mice groups for entire experimental periods. Data are expressed as means ± SD per group (**A**–**D**,**F**). *n* = 5 mice in each group (**A**–**F**). ns = not significant (*p* >0.05), ** *p* < 0.01. Scale bars = 100 um (**E**). Original magnification, 200× (**E**).

**Figure 2 nutrients-14-02195-f002:**
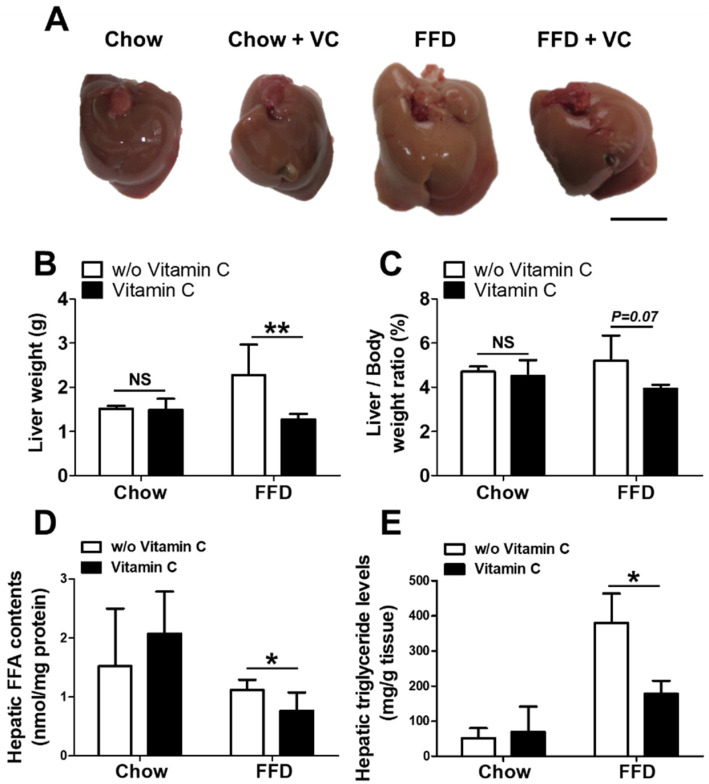
The liver phenotypes and assessment of intrahepatic fat contents. (**A**) Representative liver images of chow-diet fed and FFD-fed mice groups; (**B**) The average liver weight; and (**C**) liver/body weight ratio (%); (**D**) Hepatic free fatty acid (FFA) contents; (**E**) Hepatic triglyceride levels. Data are expressed as means ± SD per group (**B**–**E**). *n* = 5 mice in each group (**B**–**E**). ns = not significant (*p* >0.05), * *p* < 0.05, ** *p* < 0.01. Scale bars = 1 cm (**A**).

**Figure 3 nutrients-14-02195-f003:**
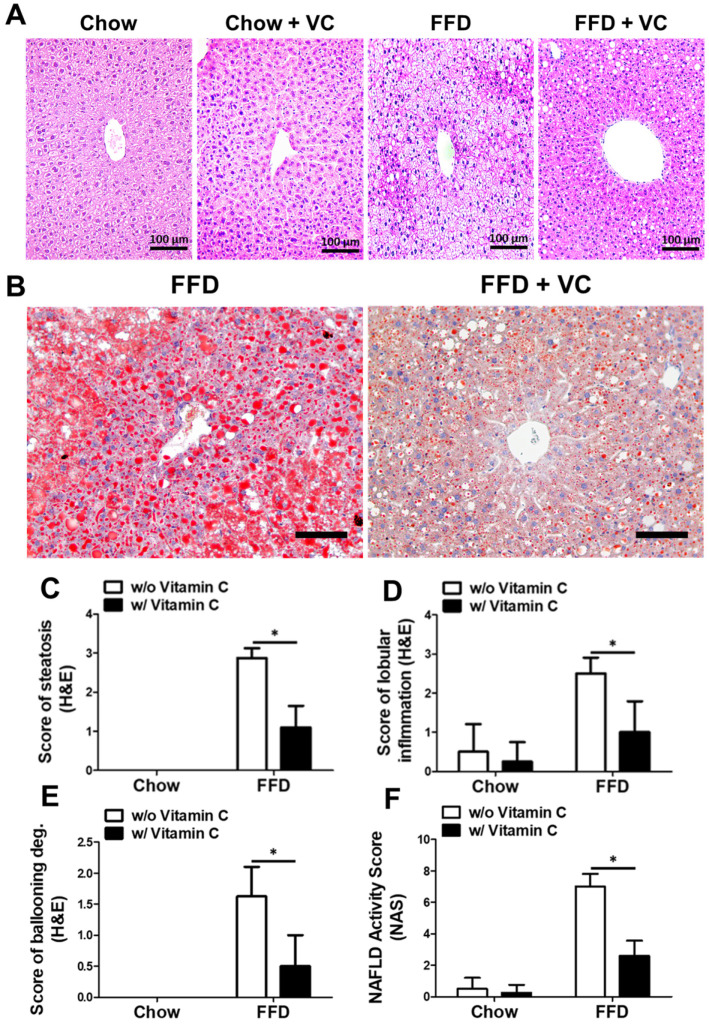
Mega-dose vitamin C treatment attenuated NAFLD progression microscopically. (**A**) Representative images of H&E staining of liver sections; (**B**) Representative images of Oil-Red O staining of frozen liver sections in FFD-fed mice groups; (**C**) Histopathological scores of steatosis; (**D**) lobular inflammation; (**E**) ballooning degeneration; and (**F**) NAFLD activity scores (NAS). Data are expressed as means ± SD per group (**C**–**F**). *n* = 5 mice in each group (**C**–**F**). * *p* < 0.05. Scale bars = 100 μm (**A**) and 200 μm (**B**). Original magnification, 200× (**A**) and 100× (**B**).

**Figure 4 nutrients-14-02195-f004:**
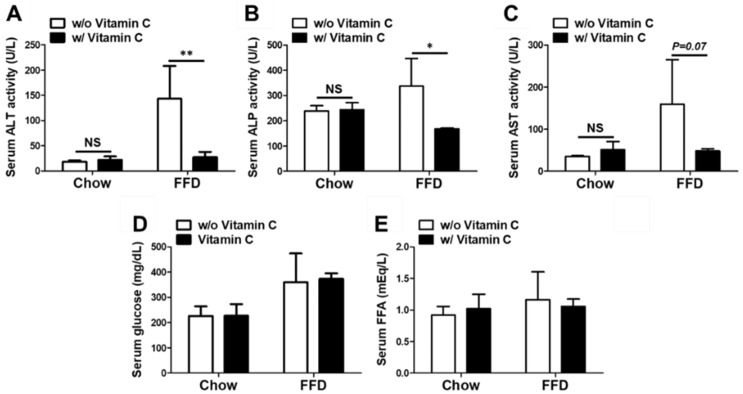
Effects of the mega-dose vitamin C treatment on the serum biochemical assays. (**A**) The serum activity levels of alanine aminotransferase (ALT); (**B**) alkaline phosphatase (ALP); and (**C**) aspartate transaminase (AST); (**D**) The serum levels of glucose; and (E) free fatty acid. Data are expressed as means ± SD per group (**A**–**E**). *n* = 5 mice in each group (**A**–**E**). * *p* < 0.05, ** *p* < 0.01.

**Figure 5 nutrients-14-02195-f005:**
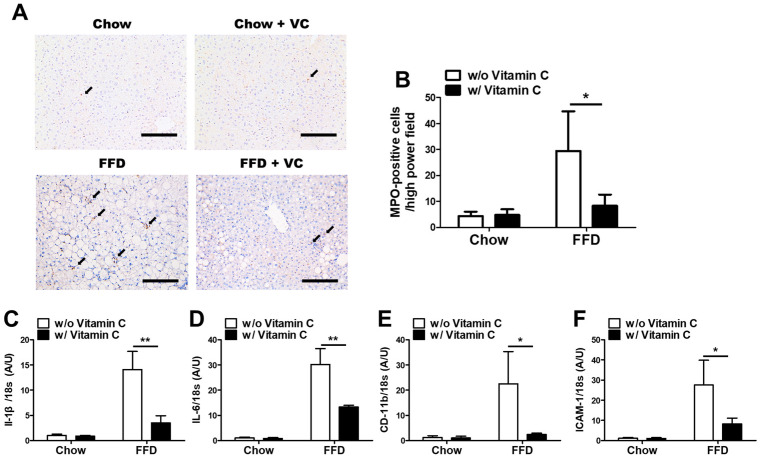
Effects of the mega-dose vitamin C treatment on the NASH-mediated inflammatory responses. (**A**) Representative images of MPO immunohistochemistry results in liver. The MPO positive cells were indicated by arrow; and (**B**) number of MPO positive cells in high power field; (**C**) Quantitative real time PCR analysis for IL-1β; (**D**) IL-6; (**E**) CD-11b; and (**F**) ICAM-1. Data were normalized to mRNA expression levels of 18s. Data are expressed as means ±SD per group (**B**–**F**). *n* = 5 mice in each group (**B**–**F**). * *p* < 0.05, ** *p* < 0.01. Scale bars = 100 μm (**A**) Original magnification, 200× (**A**).

**Figure 6 nutrients-14-02195-f006:**
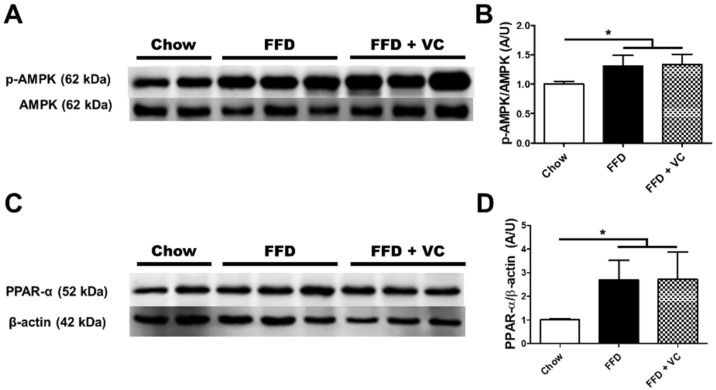
Effects of the mega-dose vitamin C treatment on catabolic pathways in whole liver tissue. (**A**) The protein expression of AMPK and phospho-AMPK; (**B**) Quantification data of AMPK phosphorylation; (**C**) The protein expression of PPAR-α and β-actin; (**D**) Quantification data of PPAR- α. Data are expressed as means ± SD per group (**B**,**D**). *n* = four to five mice in each group (**A**–**D**). * *p* < 0.05.

**Figure 7 nutrients-14-02195-f007:**
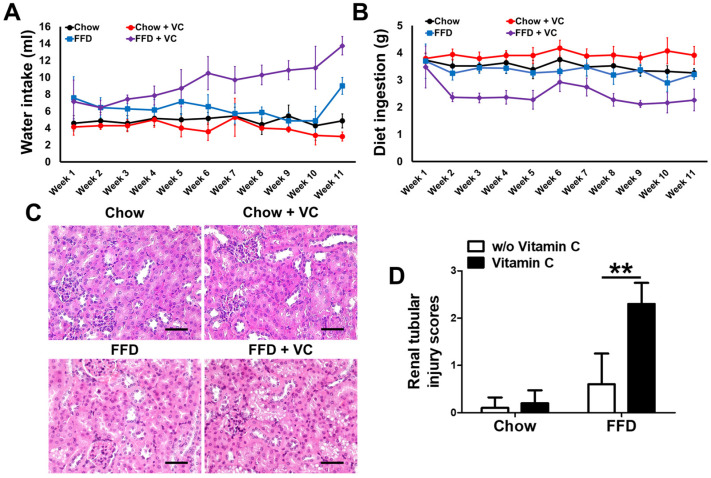
Comparison of diet ingestion, water intake and renal tubular injury. (**A**,**B**) For most of the experimental period, the FFD + VC mouse group exhibited higher water intake and lower diet food intake compared to other mice groups. (**C**) Representative H&E staining image of kidney. (**D**): Histopathological assessment of tubular damage. Data are expressed as means ± SD per group (**A**,**B**,**D**). *n* = 5 mice in each group (**A**,**B**,**D**). ** *p* < 0.01. Scale bars = 50 μm (**C**). Original magnification, 400× (**C**).

**Table 1 nutrients-14-02195-t001:** Histopathological grading system for evaluating renal tubular injury in mice.

Features	Grades/Score	Proportion of Damaged Tubules
Renal tubular injury	0	<5%
	1	5–33%
	2	33–67%
	3	67%>

**Table 2 nutrients-14-02195-t002:** Primers used in the qPCR experiment.

Target Gene	Primer Sequences (5′-3′)	
18s	Forward	ACGGAAGGGCACCACCAGGA
	Reverse	CACCACCACCCACGGAATCG
IL-1β	Forward	CTTTGAAGTTGACGGACCC
	Reverse	TGAGTGATACTGCCTGCCTG
IL-6	Forward	GCTAAGGACCAAGACCATCCAAT
	Reverse	GCTTAGGCATAACGCACTAGGTTT
CD-11b	Forward	GTGTCCGCAAGAACACCAAG
	Reverse	GGACAGGGTCTAAAGCCAGG
ICAM-1	Forward	ATTCGTTTCCGGAGAGTGTG
	Reverse	CAGCACCGTGAATGTGATCT

## Data Availability

Not applicable.

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
