# Peer review of "Mega-Dose Vitamin C Ameliorates Nonalcoholic Fatty Liver Disease in a Mouse Fast-Food Diet Model"

_nutrients, 2022, doi:10.3390/nu14112195_

Round 1
Reviewer 1 Report
Summary:
Lee and colleagues set out to assess the impact of mega-dose Vitamin C supplementation on high fat, high fructose-induced NAFLD. The authors observed a decrease in NAFLD with Vitamin C supplementation. However, there are numerous concerns that need consideration.
Major Concerns:
- Materials and Methods sections 2.7 and 2.8 are identical. There is no indication on how hepatic triglycerides were conducted.
- There is no indication in the statistical analysis section that a repeated measures ANOVA was completed for variables that were measured over time (e.g. body weight).
- Relating to line 172 – the authors state there’s increased body weight ratio in the FFD+VC group but is unclear what this is referring to. Isn’t there decreased adiposity in the FFD+VC group?
- The figure legends need to describe the figures, not the perceived outcome. For example, line 183 states the VC group significantly decreased adipocyte size. These statements should be found in the results section, not the figure legends.
- The gross images of the livers in Fig. 2A do not mean anything without a scale bar.
- It is unclear why the chow groups were omitted from several of the analyses (e.g. hepatic FFA and TG measurements).
- The samples size for each group should be noted in the figure legends.
- The authors claim there are differences between the FFD-fed vs chow-fed groups but did not use the correct statistical analysis to make these comparisons.
- pAMPK looks significantly elevated in the FFD-fed group. A greater sample size should be included for this analysis.
- A major concern is the n=5 per group for all the analysis. Another cohort of mice should be included to ensure rigor and reproducibility.
Author Response
1. Materials and Methods sections 2.7 and 2.8 are identical. There is no indication on how hepatic triglycerides were conducted.
Answer: We are sorry about this mistake, and have corrected it in the revised manuscripts.
Revised sentences
(Previous sentences, line 154-159): Blood was obtained from the caudal vena cava and collected into an eppendorf tube for 30 min at room temperature. Serum samples were centrifuged at 3000 rpm for 15 min at 4°C and stored at −80°C. The serum levels of alanine transaminase (ALT), aspartate aminotransferase (AST), alkaline phosphatase (ALP), glucose, and free fatt acid (FFA) were measured with an automated analyzer (Toshiba corporation, TBA-120FR, Tokyo, Japan).
⇓
(Revised sentences, line 206-212): Snap-frozen liver tissues were placed in isopropanol solution (Sigma, 278475, St. Louis, MO, USA) and incubated overnight at 4°C. The samples then were centrifuged at 10,000 rpm for 15 min at 4°C to aspirate the supernatants. The collected supernatants were measured with a Triglyceride L-type Kit (Wako, Osaka, Japan) and Free Fatty Acid Assay Kit (Biomax, Seoul, Korea) in accordance with the manufacturers’ instructions.
-------------------------------------------------------------------------------------
2. There is no indication in the statistical analysis section that a repeated measures ANOVA was completed for variables that were measured over time (e.g. body weight).
Answer: Thanks to the reviewer’s comments, we thoughtfully revised our statistical analysis section
Revised sentences
(Previous sentences, line 161-164): All data obtained from the experiments were expressed as the mean ± standard deviation and statistical significance among the groups was determined based on unpaired Student’s t-test, Mann-Whitney U test or Kruskal-Wallis one-way analysis of variance (Anova) on ranks.
⇓
(Revised sentences, line 214-225): All data obtained from the experiments were expressed as the mean ± standard devi-ation. D'Agostino-Pearson normality test was used to confirm the normal distribution of data. Unpaired Student’s t-test, Mann-Whitney U test or Kruskal-Wallis one-way analysis of variance (Anova) on ranks were used to examine the variables that did not show normal data distribution. Body weight, Caloric intake, water intake and diet ingestion were analyzed by two-way repeated ANOVA with the Bonferroni post-tests (time and treatment as the variables). All the statistical analysis were determined by GraphPad prism Version 5 (GraphPad Software Inc., San Diego, CA, USA) and GraphPad InStat statistical package, version 3, (GraphPad Software Inc., San Diego, CA, USA)
----------------------------------------------------------------------------------------
3. Relating to line 172 – the authors state there’s increased body weight ratio in the FFD+VC group but is unclear what this is referring to. Isn’t there decreased adiposity in the FFD+VC group?
Answer: The Increased body weight ratio data was calculated by following equation.
x = (final day body weight - initial body weight)/Initial body wieght (%)
Thus, we added increased body weight ratio to simultaneously evaluate both final day body weight and initial day body weight.
Thanks to the reviewer’s careful comments, we thoughtfully revised the sentence to explain more properly the meaning of the body weight and increased body weight results.
Revised sentences
(Previous sentences, line 171-173): The FFD + VC group exhibited significantly decreased body weight (p value = 0.0015) and increased body weight ratio (p value = 0.0045) compared with those of FFD group (Figure 1B and 1C).
⇓
(Revised sentences, line 232-238): The FFD + VC group exhibited significantly decreased body weight (p value = 0.0015) (Figure 1C). Moreover, the FFD + VC groups demonstrate almost equal increased body weight ratio (p value against to FFD group = 0.0045) with those of chow-diet fed mice groups (Figure 1D). These data indicated that the mega-dose vitamin C treatment significantly suppress the FFD-mediated body weight gain in mice.
----------------------------------------------------------------------------------------
4. The figure legends need to describe the figures, not the perceived outcome. For example, line 183 states the VC group significantly decreased adipocyte size. These statements should be found in the results section, not the figure legends.
Answer: Thanks to the reviewer’s suggestion, we thoroughly revised entire figure legends and figures.
----------------------------------------------------------------------------------------
5. The gross images of the livers in Fig. 2A do not mean anything without a scale bar.
Answer: Thanks to the editor’s constructive comments, we added the scale bar (1 cm, red circle) in Fig. 2A.
----------------------------------------------------------------------------------------
6. It is unclear why the chow groups were omitted from several of the analyses (e.g. hepatic FFA and TG measurements).
Answer: We thought that the hepatic levels of FFA and TG are major risk factors for NAFLD. So, hepatic levels of FFA and TG are measured only in the FFD-fed groups.
However, due to reviewers’ constructive comments, we measured the intrahepatic FFA and TG contents of chow-diet fed mice groups and added additional sentences in order to describe the manuscript more properly.
Moreover, we added the date of chow diet mice groups in the revised version of manuscript
Revised sentences
(Previous sentences, line 197-198): The hepatic triglyceride and FFA levels were also significantly decreased in the FFD + VC group compared to the FFD group (Figures 2C and 2D).
⇓
(Revised sentences, line 264-276): Additionally, the FFD + VC mouse group showed decreased liver weights compared to the FFD mouse group (Figure 2B and 2C). In contrast, both chow and chow + VC groups exhibited similar gross appearance and liver weights (Figures 2 A-C). In biochemical assays, the hepatic FFA, which is a main precursor for liver triglyceride synthesis, levels were significantly decreased in the FFD + VC group compared to the FFD group (Figure 2C). The hepatic triglyceride levels were significantly increased in all FFD-fed mice groups compared to chow-diet fed groups suggesting FFD-mediated NAFLD progression. However, the FFD + VC group exhibited notably decreased liver triglyceride levels compared to FFD group (Figure 2D). These data indicated that the vitamin C treatment selectively attenuated fast-food-diet mediated fat accumulation in liver.
----------------------------------------------------------------------------------------
7. The samples size for each group should be noted in the figure legends.
Answer: Thanks to the reviewer’s comments, we thoughtfully revised all figure legends and added samples size for each group.
----------------------------------------------------------------------------------------
8. The authors claim there are differences between the FFD-fed vs chow-fed groups but did not use the correct statistical analysis to make these comparisons
Answer: Thanks to the reviewer’s constructive comments, we thoughtfully revised the statistical analysis section.
----------------------------------------------------------------------------------------
9. pAMPK looks significantly elevated in the FFD-fed group. A greater sample size should be included for this analysis.
Answer: As reviewers’ insightful comments, we performed additional western blot analysis to increase sample size (n = 4 to 5 per each group). As a result, we confirm that the phosphorylation of AMPK was significantly increased in FFD-fed mice groups.
Therefore, we carefully revised the AMPK quantification figures
----------------------------------------------------------------------------------------
10. A major concern is the n=5 per group for all the analysis. Another cohort of mice should be included to ensure rigor and reproducibility.
Answer: We totally agree with the reviewers' comments. It is important to assign sufficient number of mice in each experimental group. At first, we planned to use at least 8 ~ 10 mice per groups for rigor and reproducibility of data. However, in present study, there were no procedures were performed on animals except ad-libitum diet ingestion, and thus, it is expected that the mortality rate of mouse is very low. Thus, our animal ethics committee (KNU-IACUC) suggests reducing the total number of animal use.
As a result, each mouse groups were consisted of 5 animals. Additionally, the standard deviation in the present study were generally low despite of relatively small group size. Additionally, we also found that there were many already published research which have same animal group size with the present study. [1-3]
Reference
[1] LEE, Young-Sil, et al. Tetragonia tetragonoides (Pall.) Kuntze (New Zealand Spinach) prevents obesity and hyperuricemia in high-fat diet-induced obese mice. Nutrients, 2018, 10.8: 1087.
[2] UPRETI, Deepa, et al. Oral Administration of Water Extract from Euglena gracilis Alters the Intestinal Microbiota and Prevents Lung Carcinoma Growth in Mice. Nutrients, 2022, 14.3: 678.
[3] SATO, Ban, et al. Suppressive Role of Lactoferrin in Overweight-Related Female Fertility Problems. Nutrients, 2022, 14.5: 938.
----------------------------------------------------------------------------------------

Reviewer 2 Report
- The details of the two groups were not mentioned in the abstract to be found out.
- Please write the statistical analyses in detail, that is, which tests were used for what aims?
- Please write limitations in the discussion.
- Please write your conclusions in a separate paragraph.
Author Response
1. The details of the two groups were not mentioned in the abstract to be found out.
Answer: Thanks to the reviewer’s valuable comments, we added more detailed description about each mice group in abstract parts
Revised sentences
(Previous sentences, line 20-34): In previous studies, the increasing clinical importance of nonalcoholic fatty liver disease (NAFLD) has been recognized. However, the specific therapeutic strategies or drugs have not been discovered. Vitamin C is a water-soluble antioxidant and is a cofactor in many important biosynthesis pathways. Recently, many researchers have reported that mega-dose vitamin C treatment had positive effects on various diseases. However, the precise relationship between mega-dose vitamin C and NAFLD has not been completely elucidated. This study has been designed to discover the effects of mega-dose vitamin C on the progression of NAFLD. Thirteen-week-old wild-type C57BL6 mice were fed a high-fat and high-fructose diet (fast-food diet) ad libitum for 11 weeks. Vitamin C was administered in the drinking water (1.5 g/L). In this study, 11 weeks of mega-dose vitamin C treatment significantly suppressed the development of nonalcoholic steatohepatitis (NASH) independently of the catabolic process. Vitamin C supplements in fast-food diet fed mice significantly decreased diet ingestion and increased water intake. Histopathological analysis revealed that the mice fed a fast-food diet with vitamin C water had a mild renal injury suggesting osmotic nephrosis due to fructose-mediated purine derivatives. These data suggest that mega-dose vitamin C treatment suppresses high-fructose-diet-mediated NAFLD progression by decreasing diet ingestion and increasing water intake
⇓
(Revised sentences, line 18-34): In previous studies, the increasing clinical importance of nonalcoholic fatty liver disease (NAFLD) has been recognized. However, the specific therapeutic strategies or drugs have not been discovered. Vitamin C is a water-soluble antioxidant and is a cofactor in many important biosynthesis pathways. Recently, many researchers have reported that mega-dose vitamin C treatment had positive effects on various diseases. However, the precise relationship between mega-dose vitamin C and NAFLD has not been completely elucidated. This study has been designed to discover the effects of mega-dose vitamin C on the progression of NAFLD. Twelve-week-old wild-type C57BL6 mice were fed chow diets and high-fat and high-fructose diet (fast-food diet) ad libitum for 11 weeks with or without of vitamin C treatment. Vitamin C was administered in the drinking water (1.5 g/L). In this study, 11 weeks of mega-dose vitamin C treatment significantly suppressed the development of nonalcoholic steatohepatitis (NASH) independently of the catabolic process. Vitamin C supplements in fast-food diet fed mice significantly decreased diet ingestion and increased water intake. Histopathological analysis revealed that the mice fed a fast-food diet with vitamin C water had a mild renal injury suggesting osmotic nephrosis due to fructose-mediated purine derivatives. These data suggest that mega-dose vitamin C treatment suppresses high-fructose-diet-mediated NAFLD progression by decreasing diet ingestion and increasing water intake.
----------------------------------------------------------------------------------------
2. Please write the statistical analyses in detail, that is, which tests were used for what aims?
Answer: Thanks to the reviewer’s valuable comments, we thoughtfully revised our statistical analysis section
Revised sentences
(Previous sentences, line 161-164): All data obtained from the experiments were expressed as the mean ± standard deviation and statistical significance among the groups was determined based on unpaired Student’s t-test, Mann-Whitney U test or Kruskal-Wallis one-way analysis of variance (Anova) on ranks.
⇓
(Revised sentences, line 214-225): All data obtained from the experiments were expressed as the mean ± standard devi-ation. D'Agostino-Pearson normality test was used to confirm the normal distribution of data. Unpaired Student’s t-test, Mann-Whitney U test or Kruskal-Wallis one-way analysis of variance (Anova) on ranks were used to examine the variables that did not show normal data distribution. Body weight, Caloric intake, water intake and diet ingestion were analyzed by two-way repeated ANOVA with the Bonferroni post-tests (time and treatment as the variables). All the statistical analysis were determined by GraphPad prism Version 5 (GraphPad Software Inc., San Diego, CA, USA) and GraphPad InStat statistical package, version 3, (GraphPad Software Inc., San Diego, CA, USA)
----------------------------------------------------------------------------------------
3. Please write limitations in the discussion.
Answer: Thanks to the reviewer’s helpful comments, we added limitations part in the discussion
(Newly added sentences, line 488-498): In the present study, we thoroughly investigate the effects of mega-dose vitamin C treatment in NAFLD progression. However, there were some limitations of our current experiments. First, we suggested that the mega-dose vitamin C treatment accelerated degradations of the FFD-mediated purine derivatives in mice. However, the precise mechanisms of mega-dose vitamin C treatment in uricase activities are still unknown. Seconds, all results observed in the present study come from animal experiments, and thus it can’t be directly applied to human beings because of the species differences. Given that, it seems that furthered studies are needed to confirm and expand these findings.
----------------------------------------------------------------------------------------
4. Please write your conclusions in a separate paragraph.
Answer: Thanks to the reviewer’s helpful comments, we newly added the conclusions in the revised version of manuscript
(Newly added sentences, line 500-508): In summary, we have demonstrated that the mega-dose treatment of vitamin C significantly attenuated FFD-mediated NAFLD by decreasing diet ingestion and increasing water intake. These data also suggested that the mega-dose vitamin C treatment facilitated the excretion of FFD-mediated purine derivatives in mice. Taken together, our study showed the beneficial effects of mega-dose vitamin C treatment in NAFLD, and we anticipated that the results from the present study would lead to a better understanding of the relationship between mega-dose vitamin C treatment and NAFLD.

Reviewer 3 Report
In my opinion the manuscript entitled „Mega-dose Vitamin C ameliorates Nonalcoholic Fatty Liver Disease in a Mouse Fast-food Diet Model” requires significant corrections as Authors did not explain the aim of this study in a sufficiently convincing way. Moreover, the text needs to be supplemented with information on the research methodology. The discussion requires major corrections, and the conclusions of the study are also missing.
Introduction
- Authors incorrectly gave the definition of NAFLD, indicating alcohol consumption as the cause of fatty liver.
- Line 46 - it is not true that there is no treatment strategy for NAFLD. Please present the current state of knowledge in this field.
- Line 52 - this issue is debatable, please explain whether excess vitamin C is in fact unequivocally safe?
- Line 55 - please specify in which studies and in what doses and whether there were side effects of these therapies. Please separate animal studies from human studies.
- Line 53-60 - the research is described in a laconic way. Authors do not adequately justify the purpose of their study. This section needs a major revision.
Materials and Methods
- Each table must have a title and description.
- No full stop at the end of the sentence.
- Please provide a study scheme.
- How was the vitamin C doses determined?
- Did Authors get approval from the ethics committee?
- How did Authors verify the normal distribution of data? What tests they used?
- Subsections 2.7 and 2.8 are the same. Please provide the method of parameter determination
Results
- Figure 1 is illegible.
- The results of other studies are not reported in this section.
- Please remove the aim of study from this section.
- Please provide the results for the control group.
Discussion - this part requires a major revision.
- Please remove the references to the tables.
- Line 321-354 - There is no reference to the results obtained by other authors.
- Authors repeat the information from the Introduction section.
- Line 356-362 - What relationship with NAFLD? Please concentrate on NAFLD.
- Line 369-370 - Why Authors extrapolate animal studies to humans? Please explain it and state what amounts of vitamin C NAFLD patients would have to use to achieve a therapeutic effect.
- Line 372-386 - What relationship with NAFLD? Please concentrate on NAFLD.
- please explain the usefulness of the results obtained. Should patients with NAFLD use mega-doses of vitamin C?
- Please also indicate the limitations of this study.
Conclusions
- Please complete the conclusions.
References
- The way of citation needs to be improved.
Author Response
Introduction
1. Authors incorrectly gave the definition of NAFLD, indicating alcohol consumption as the cause of fatty liver.
Answer: We are sorry about this mistake, and have corrected it in the revised manuscripts
Revised sentences
(Previous sentences, line 44-45): It is generally defined as the accumulation of hepatic triglycerides in excess of 5% of the liver weight with alcoholic intake, viral infection, and drug treatments [1,2].
⇓
(Revised sentences, line 45-46): It is generally defined as the accumulation of hepatic triglycerides in excess of 5% of the liver weight without alcoholic intake, viral infection, and drug treatments [1,2].
----------------------------------------------------------------------------------------
2. Line 46 - it is not true that there is no treatment strategy for NAFLD. Please present the current state of knowledge in this field.
Answer: We agree with the reviewers' comments, and properly revised the sentences
Revised sentences
(Previous sentences, line 45-46): Although the global burden of NAFLD is increasing, specific drugs and therapeutic strat-egies to treat NAFLD have not been found
⇓
(Revised sentences, line 57-61): Because the global burden of NAFLD is constantly increasing, there were many research about the specific drugs and therapeutic strategies to treat NAFLD. However, the precise pathogenesis and specific therapeutic strategies of NAFLD have not been fully understood yet, and additional research is urgently needed.
----------------------------------------------------------------------------------------
3. Line 52 - this issue is debatable, please explain whether excess vitamin C is in fact unequivocally safe?
Answer: We agree with the reviewers' comments that there were controversies of mega-dose vitamin C treatment. Vitamin C is a highly-water soluble nutrient, and it is generally known that the vitamin C has no toxicity even on high dose intakes in humans. [1]
However, some researchers also reported that the high-dose intake of vitamin C might cause various side effects in human beings. [2] Thus, we planned to conduct pilot experiment with wild-type mice to confirm safety of long-term megadose vitamin C treatment.
Before the current study, vitamin C at the same dose used in the present study (1.5g/L in filtered tap water) was provide to wild-type mice for 70 weeks, and all vitamin C treated mice exhibited no evidence of parenchymal organ injuries based on the histopathologic examination. Given that, we can confirm that the safety of mega-dose vitamin C treatment in mice, and added the sentences about these limitation in the revised version of manuscript.
However, when considering the differences between human and mouse in metabolic processes, it seems that further studies in humans also need to be conducted to confirm these data.
Revised sentences
(Newly added sentences, line 112-115): Additionally, Twelve-week-old male C57BL/6N mice (n = 5) were housed at same condition in the described above. These mice were fed by chow diet and vitamin C contained filtered tap water for 70 weeks to confirm safety of long-term mega-dose vitamin C treatment.
(Newly added sentences, line 120): The mouse tissue samples were collected at 11 and 70 weeks and fixed in 10% neutral buffered formalin
(Newly added sentences, line 467-476): In present study, the FFD + VC mice exhibited mild kidney injuries compared than other mice groups (Figure 7). Additionally, there were few researches reported that the mega-dose vitamin C might have adverse effects [27]. Therefore, at first, we performed animal experiments to confirm the safety of mega-dose vitamin C treatment in animals. As a result, the wild-type C57BL/6 mice treated by vitamin C at the same dose used in the present study for 70 weeks did not have any significant liver and kidney damage (Figure S1). Consequently, we can confirm that the safety of mega-dose vitamin C treatment.
(Newly added sentences, line 488-498): In the present study, we thoroughly investigate the effects of mega-dose vitamin C treatment in NAFLD progression. However, there were some limitations of our current experiments. First, we suggested that the mega-dose vitamin C treatment accelerated degradations of the FFD-mediated purine derivatives in mice. However, the precise mechanisms of mega-dose vitamin C treatment in uricase activities are still unknown. Seconds, all results observed in the present study come from animal experiments, and thus it can’t be directly applied to human beings because of the species differences. Given that, it seems that furthered studies are needed to confirm and expand these findings.
Refererence
[1] SCHLUETER, Amanda K.; JOHNSTON, Carol S. Vitamin C: overview and update. Journal of Evidence-Based Complementary & Alternative Medicine, 2011, 16.1: 49-57.
[2] PADAYATTY, Sebastian J., et al. Vitamin C: intravenous use by complementary and alternative medicine practitioners and adverse effects. PloS one, 2010, 5.7: e11414.
----------------------------------------------------------------------------------------
4. Line 55 - please specify in which studies and in what doses and whether there were side effects of these therapies. Please separate animal studies from human studies.
5. Line 53-60 - the research is described in a laconic way. Authors do not adequately justify the purpose of their study. This section needs a major revision.
Answer: Thanks to the reviewer’s constructive suggestion, we carefully revised the sentences to explain this point more clearly.
Revised sentences
(Previous sentences, line 53-60): Mega-dose vitamin C usage, characterized by doses over the recommended daily amount, is considered a potential therapy for many diseases, and many researchers have reported that mega-dose vitamin C treatment demonstrates a high therapeutic potential for various diseases [8]. In previous studies, mega-dose vitamin C drastically attenuated multi-organ dysfunction in a sheep model of sepsis caused by gram-negative bacteria, and it is also exhibited potential therapeutic effects on COVID-19 in human patients [9]. Researchers have also suggested that mega-dose vitamin C might inhibit growth of tumor cells by enhancing human immune responses [10]. Previous studies have shown mega-dose treatment with vitamin C effectively reduces serum uric acid levels in humans and has positive effects in uric-acid related disease [11,12]. Due to this research, it is hypothesized that mega-dose treatment with vitamin C might have beneficial effects on other metabolic disorders including NAFLD. Mega-dose vitamin C treatment for NAFLD has not been studied. Therefore, we planned animal experiments to study the effects of mega-dose vitamin C on NAFLD.
⇓
(Revised sentences, line 70-91): According to previous studies, mega-dose vitamin C usage, characterized by doses over the recommended daily amount, is considered a potential therapy for many diseases, and many researchers have reported that mega-dose vitamin C treatment demonstrates a high therapeutic potential for various diseases. In previous animal model study, mega-dose vitamin C treatment (Intravenous, 150 g per 40 kg over 7h) drastically attenuated multi-organ dysfunction in a sheep model of sepsis caused by gram-negative bacteria [8]. In humans, mega-dose vitamin C treatment (Intravenous, 60 g) also exhibited potential therapeutic effects on COVID-19 patients [8]. Other researchers have also suggested that mega-dose vitamin C (Intravenous, 5-45g per day and/or Oral, 5-20g per day) might inhibit-it growth of tumor cells by enhancing human immune responses [9]. Other researchers also reported that the negative correlation between megadose vitamin C treatments and attenuated uric acid related diseases in humans [10, 11]. Moreover, any adverse effects of mega-dose vitamin C treatment were not reported in any of these studies [7-11]. Given that, it is hypothesized that mega-dose treatment with vitamin C might have beneficial effects on metabolic disorders. However, the precise mechanisms between mega-dose vitamin C treatment and NAFLD pathogenesis have not yet been completely studied yet. Therefore, we planned animal experiments to study the effects of mega-dose vitamin C on NAFLD.
----------------------------------------------------------------------------------------
Materials and Methods
1. Each table must have a title and description.
Answer: Thanks to the reviewer’s helpful comments, we added title to each table
Newly added sentences
(Line 130-131): Histopathological grading system for evaluating renal tubular injury in mice
(Line 195): Primers used in the qPCR experiment
----------------------------------------------------------------------------------------
2. No full stop at the end of the sentence.
Answer: Thanks to the reviewer’s constructive comments, we have revised the manuscript according to reviewer’s comments
----------------------------------------------------------------------------------------
3. Please provide a study scheme.
Answer: Thanks to the reviewer’s constructive comments, we added schematic diagram of present study in Figure 1 and added figure description.
Newly added sentences
(Line 247-248): (A): Schematic diagram of the experimental protocol used to assess the effect of mega-dose vitamin C treatment on NAFLD
----------------------------------------------------------------------------------------
4. How was the vitamin C doses determined?
Answer: The vitamin C doses in present study were mainly determined by previous studies. According to previous studies, Kondo and Ishigami et al. reported that vitamin C deficient transgenic mice treated with 1.5g/L of vitamin C contained water exhibited that similar serum vitamin C concentration compared to wild-type mice [ 1 ].
Moreover, previous experiment also reported the wild-type mice can synthesis vitamin C endogenously [ 2 ]. Therefore, wild-type mice can maintain normal serum vitamin C concentration without additional supplements of it.
Taken together, we suggested that the 1.5g/L of vitamin C treatment on wild-type mice is a suitable method for investigating the effects of mega-dose vitamin C treatment.
Subsequently, in present study, the FFD + VC group also demonstrated significantly attenuated NAFLD progression compared than FFD group
Reference
[ 1 ]; KONDO, Yoshitaka, et al. Vitamin C depletion increases superoxide generation in brains of SMP30/GNL knockout mice. Biochemical and biophysical research communications, 2008, 377.1: 291-296
[2] DONG, Wei, et al. Multiple genome analyses reveal key genes in Vitamin C and Vitamin D synthesis and transport pathways are shared. Scientific reports, 2019, 9.1: 1-12.
----------------------------------------------------------------------------------------
5.Did Authors get approval from the ethics committee?
Answer: Thanks to the reviewer’s constructive comments, we moved the sentences about our ethics approval to material and methods section
Additionally, we also added new ethics approval number for our supplementary figure (Figure S1)
Revised sentences
(Previous sentences, line 414-415): Institutional Review Board Statement: Animal use and procedures were approved by the Kyungpook National University Institutional Animal Care and Use Committee (IACUC, approval number 2017-0112 and 2021-0175)
⇓
(Revised sentences, line 115-118): Animal use and procedures were approved by the Kyungpook National University Institutional Animal Care and Use Committee (IACUC, approval number 2017-0112, 2021-0175 and 2021-0176)
----------------------------------------------------------------------------------------
6. How did Authors verify the normal distribution of data? What tests they used?
Answer: Thanks to the reviewer’s comments, we thoughtfully revised our statistical analysis section
Revised sentences
(Previous sentences, line 161-164): All data obtained from the experiments were expressed as the mean ± standard deviation and statistical significance among the groups was determined based on unpaired Student’s t-test, Mann-Whitney U test or Kruskal-Wallis one-way analysis of variance (Anova) on ranks.
⇓
(Revised sentences, line 214-225): All data obtained from the experiments were expressed as the mean ± standard devi-ation. D'Agostino-Pearson normality test was used to confirm the normal distribution of data. Unpaired Student’s t-test, Mann-Whitney U test or Kruskal-Wallis one-way analysis of variance (Anova) on ranks were used to examine the variables that did not show normal data distribution. Body weight, Caloric intake, water intake and diet ingestion were analyzed by two-way repeated ANOVA with the Bonferroni post-tests (time and treatment as the variables). All the statistical analysis were determined by GraphPad prism Version 5 (GraphPad Software Inc., San Diego, CA, USA) and GraphPad InStat statistical package, version 3, (GraphPad Software Inc., San Diego, CA, USA).
----------------------------------------------------------------------------------------
7. Subsections 2.7 and 2.8 are the same. Please provide the method of parameter determination
Answer: We are sorry about this mistake, and have corrected it in the revised manuscripts
Revised sentences
(Previous sentences, line 154-159): Blood was obtained from the caudal vena cava and collected into an eppendorf tube for 30 min at room temperature. Serum samples were centrifuged at 3000 rpm for 15 min at 4°C and stored at −80°C. The serum levels of alanine transaminase (ALT), aspartate aminotransferase (AST), alkaline phosphatase (ALP), glucose, and free fatt acid (FFA) were measured with an automated analyzer (Toshiba corporation, TBA-120FR, Tokyo, Japan).
⇓
(Revised sentences, line 206-212): Snap-frozen liver tissues were placed in isopropanol solution (Sigma, 278475, St. Louis, MO, USA) and incubated overnight at 4°C. The samples then were centrifuged at 10,000 rpm for 15 min at 4°C to aspirate the supernatants. The collected supernatants were measured with a Triglyceride L-type Kit (Wako, Osaka, Japan) and Free Fatty Acid Assay Kit (Biomax, Seoul, Korea) in accordance with the manufacturers’ instructions.
----------------------------------------------------------------------------------------
Results
1. Figure 1 is illegible.
Answer: As the reviewer’s pointed out, we totally revised the Figure 1 to explain the results more properly.
Revised sentences
(Previous sentences, line 168-177): To evaluate the precise effects of vitamin C in diet-mediated NAFLD, the mice were fed a chow diet or a fast food diet with or without vitamin C for 11 weeks. The FFD + VC group exhibited lower body weight gain compared with the FFD group for the experimental period (Figure 1A). The FFD + VC group exhibited significantly decreased body weight (p value = 0.0015) and increased body weight ratio (p value = 0.0045) compared with those of FFD group (Figure 1B and 1C). Microscopically, the FFD + VC group demonstrated significantly decreased average adipocyte size compared than FFD group (Figure 1D and 1E). However, the caloric intake per body weight was equal in all FFD-fed groups for the experimental period (Figure 1). These results suggest that mega-dose vitamin C treatment markedly decreased body weight gain in the FFD mice groups
(Previous sentences, line 179-186): Comparison of body weight between each mouse group. (A): For most of the experimental pe-riod, the FFD + VC group exhibited significantly lower average body compared than FFD group. (B and C): Consistently, the FFD + VC group exhibited significantly decreased body weight and increased body weight ratio. (D and E): Relative size and representative images of hematoxylin and eosin (H&E) staining of the adipocytes in FFD-fed mouse groups. The FFD + VC group demonstrated significantly decreased average adipocyte size compared than FFD group. (F): All FFD-diet fed mice exhibited almost equal calories intake per body weight despite of phenotypi-cal differences. *P < 0.05, **P < 0.01. Scale bars = 100 um (E). Original magnification, X 200 (E).
⇓
(Revised sentences, line 229-244): The mice were fed a chow diet or a fast-food diet with or without vitamin C for 11 weeks. (Figure 1A) The FFD + VC group exhibited lower body weight gain compared with the FFD group for the experimental period (Figure 1B). The FFD + VC group exhibited significantly decreased body weight (p value = 0.0015) (Figure 1C). Moreover, the FFD + VC groups demonstrate almost equal increased body weight ratio (p value against to FFD group = 0.0045) with those of chow-diet fed mice groups (Figure 1D). These data indicated that the mega-dose vitamin C treatment significantly suppress the FFD-mediated body weight gain in mice. Microscopically, the FFD + VC group demonstrated significantly decreased average adipocyte size compared than FFD group (Figure 1E and 1F). However, the caloric intake per body weight was equal in all FFD-fed groups for the experimental period (Figure 1G). These results suggest that mega-dose vitamin C treatment markedly decreased body weight gain in the FFD mice groups
(Revised sentences, line 246-255): Comparison of body weight and calories intake between each mouse group. (A): Schematic diagram of the experimental protocol used to assess the effect of mega-dose vitamin C treatment on NAFLD (B): The body weight data of whole mice groups for entire experimental periods. (C): The body weight of final day; and (D) The increased body weight ratio (%) (E): Representative images of hematoxylin and eosin (H&E) staining; and (F) relative size of adipocytes. (G): The calories intake per body weight levels in whole mice groups for entire experimental periods. Data are expressed as means ±SD per group (A-D and F). n=5 mice in each group (A-F). *P < 0.05, **P < 0.01. Scale bars = 100 um (E). Original magnification, X 200 (E)
----------------------------------------------------------------------------------------
2. The results of other studies are not reported in this section
Answer: As the reviewer’s pointed out, we removed the results of other studies in result part
Revised sentences
(Previous sentences, line 250-255): In previous studies, prolonged high-carbohydrate and high-fat diets induce NASH development not only triglyceride accumulation in hepatocytes but also through infiltration of inflammatory cells into the liver parenchyma. [20] We hypothesized that vitamin C might block the migration of cytotoxic inflammatory cells into the liver parenchyma. We hypothesized that vitamin C might block the migration of cytotoxic inflammatory cells into the liver parenchyma. We performed immunohistochemistry using anti-MPO antibodies which are a major marker of neutrophils.
(Previous sentences, line 298-306): In a previous study excessive intake of fructose increased purine derivatives including uric acid, allantoin and urea levels in the serum, and these purine derivatives are mainly excreted by the renal system in rodents [21]. Urea is a water-soluble end product of purine degradation. Since these solutes can cause a osmotic diuresis and osmotic nephrosis, we hypothesized that the FFD + VC group had increased excretion of purine derivatives including uric acid, allantoin, and urea in the urine compared with those in the FFD group [22]. This hypothesis is consistent with the microscopic findings in the FFD + VC mice with prominent cytoplasmic vacuoles and swelling in cells of the proximal renal tubule suggesting mild osmotic nephrosis. (Figure 7C and 7D). [23]
⇓
(Revised sentences, line 335-338): Next, we performed immunohistochemistry using anti-MPO antibodies which are a major marker of neutrophils. In immunohistochemistry staining, the FFD group showed many neutrophils
(Revised sentences, line 388-392): Following histopathologic assessment revealed mild renal tubular injuries in FFD + VC group compared to FFD group, suggesting increased osmotic pressures in FFD + VC mice than FFD group (Figure 7C and 7D).
----------------------------------------------------------------------------------------
3. Please remove the aim of study from this section.
Answer: As the reviewer’s pointed out, we thoughtfully revised the sentences in result parts
----------------------------------------------------------------------------------------
4. Please provide the results for the control group.
Answer: As the reviewer’s pointed out, we added the results of control group
----------------------------------------------------------------------------------------
Discussion
1. Please remove the references to the tables.
Answer: Thanks to the reviewer’s constructive comments, we have revised the manuscript according to reviewer’s comments
----------------------------------------------------------------------------------------
2. Line 321-354 - There is no reference to the results obtained by other authors.
Answer: Thanks to the reviewer’s constructive comments, we added references about the results by other authors
(Newly added sentences, Line 418-422): In previous studies, prolonged high-carbohydrate and high-fat diets induce NASH development not only triglyceride accumulation in hepatocytes but also through infiltration of inflammatory cells into the liver parenchyma. [19]
(Newly added sentences, Line 434-438): According to previous study, excessive intake of fructose-rich diet increased the serum levels of purine derivatives including uric acid, allantoin and urea, and these purine derivatives are mainly excreted by the renal system in rodents [20,21].
(Newly added sentences, Line 438-442): Since these solutes can cause an osmotic diuresis and osmotic nephrosis, we hypothesized that the FFD + VC group had increased excretion of purine derivatives including uric acid, allantoin, and urea in the urine compared with those in the FFD group [22].
(Newly added sentences, Line 443-445): the FFD + VC mice with prominent cytoplasmic vacuoles and swelling in cells of the proximal renal tubule suggesting mild osmotic nephrosis. (Figure 7C and 7D) [23].
----------------------------------------------------------------------------------------
3. Authors repeat the information from the Introduction section.
4. Line 356-362 - What relationship with NAFLD? Please concentrate on NAFLD.
Answer: We agree with the reviewer’s suggestion; these sentences were repeated and overstated and the relevant text has been removed in revised manuscript
(Removed sentences, line 350-355): NAFLD not only induce fatty liver disease but provoke the end-stage liver diseases such as cirrhosis and hepatocellular carcinoma [25]. Recently, NAFLD has become a one of most common chronic liver diseases because of the global spread of Western-style diets and lack of physical activity [3]. There have been many studies to find novel therapeutic agents for treating NAFLD [26]. However, specific therapies for NAFLD have not been found, and additional research is still needed.
(Removed sentences, line 356-362): Vitamin C is an essential nutrient involved in many biosynthesis pathways and also plays a crucial role in reducing oxidative stress in cells [27]. In previous studies, vitamin C demonstrated efficacy in various diseases. L. Ran et al. found that vitamin C had potential in treating the common cold by reduced symptoms [28]. N. Zhu et al. showed that vitamin C had protective effects on cardiovascular diseases due to its anti-inflammatory properties [29]. Additionally, J. Kocot et al. suggested that the vitamin C treatment demonstrated a favorable result in many neurodegenerative diseases
----------------------------------------------------------------------------------------
5. Line 369-370 - Why Authors extrapolate animal studies to humans? Please explain it and state what amounts of vitamin C NAFLD patients would have to use to achieve a therapeutic effect.
Answer: According to previous study, the high-fructose and high-fat diet (fast-food diet, FFD) mediated animal model of NAFLD not only effective in producing NAFLD compared to simple high-fat diet treatment but also have many similarities with human NAFLD pathogenesis [1, 2].
Therefore, in present study, we investigated the effects of mega-dose vitamin C treatment of FFD -fed mice model and revealed that the mega-dose vitamin C treatment on mice successfully attenuated the FFD-mediated NAFLD progressions.
When considering that the FFD-mediated NAFLD has a similar pathogenesis to human NAFLD, we expect that the mega-dose vitamin C treatment might also have a positive effect on human NAFLD.
In present study, the average vitamin C oral intake dose of FFD + VC mice was approximately 750 mg/kg in daily. When considering the human equivalent dose (HED) methods, the human equivalent dose corresponding 750 mg/kg in mice is 60 mg/kg (assuming 60 kg of body weigh). [3] Taken together, it is suggested that the 3-5 grams per day, oral vitamin C intake is estimated human dose according to the present study. However, when considering the differences between human and mouse, it would be difficult to directly apply it to human, and further research is needed to confirm these data
Reference
[ 1 ] ; JENSEN, Thomas, et al. Fructose and sugar: A major mediator of non-alcoholic fatty liver disease. Journal of hepatology, 2018, 68.5: 1063-1075.
[ 2 ] ; ASGHARPOUR, Amon, et al. A diet-induced animal model of non-alcoholic fatty liver disease and hepatocellular cancer. Journal of hepatology, 2016, 65.3: 579-588.
[ 3 ] : SHIN, Jang-Woo; SEOL, In-Chan; SON, Chang-Gue. Interpretation of animal dose and human equivalent dose for drug development. The Journal of Korean Medicine, 2010, 31.3: 1-7.
----------------------------------------------------------------------------------------
6. Line 372-386 - What relationship with NAFLD? Please concentrate on NAFLD.
Answer: We agree with the reviewer’s opinion, these sentences did not explain effectively the result in present study, and we thoughtfully revised these parts in revised version of manuscript.
Revised sentences
(Previous sentences, line 372-386): In prior research, excessive ingestion of fructose (e.g., a fast food diet) caused increased levels of uric acid via fructose-mediated purine degradation in cells [31]. In humans, uric acid is mainly cleared by the renal system, and the remaining uric acid is degraded by intestinal uricolysis [32]. Unfortunately, uric acid has a lower water-solubility, and excess uric acid in serum can transform into urate crystals [33]. These urate crystals can cause gout with urate crystal deposition in joints and a chronic inflammatory response [34]. Most mammals, with the exception of primates, can convert uric acid to allantoin and urea which are more water-soluble metabolites than uric acid [35]. Therefore, mice have a relatively lower risk of gout than primates.
There are many hypotheses as to why primates have lost uricase synthesis activity despite the increasing risk of the urate mediated diseases such as hyperuricemia and gout. Uric acid is a strong antioxidant and despite its high risks of urate related disease, since primates do not synthesis vitamin C it may be that uricase mutation may have occurred as a means to compensate for loss of the antioxidant activity of vitamin C in primates. [36,37] Therefore, we hypothesis that vitamin C might replace the antioxidant effects of uric acid and facilitate the renal excretion of it.
⇓
(Revised sentences, line 457-466): Most of mammals, except for primates, excreted excessive purine derivatives to allantoin and urea which are more water-soluble solutes than uric acid via enzymatic reactions by uricase [24]. Therefore, it assumed that mega-dose vitamin C treatment enhance uricase activity in FFD-fed mice. Interestingly, there have been many studies using uricase or uricase recombinant to treat uric acid related disease in humans [25, 26]. When considering this previous research and the data in the present study, it is suggested that in combination with uricase treatment and mega-dose vitamin C treatment might have a synergistic effect in treating uric acid related disease in human.
----------------------------------------------------------------------------------------
7. please explain the usefulness of the results obtained. Should patients with NAFLD use mega-doses of vitamin C?
Answer: In present study, the megadose vitamin C treatment notably ameliorated FFD-mediated NAFLD. Additionally, we also confirm the long-term safety of megadose vitamin C treatment. When considering that the FFD-mediated NAFLD has similar pathogenesis with human NAFLD [ 1 ], we assumed that the megadose-vitamin C treatment might have a positive effect on human NAFLD.
Thanks to the reviewer’s helpful comments, we newly added conclusion parts in revised version of manuscript to explain usefulness of present study.
(Newly added sentences, line 500-508): In summary, we have demonstrated that the mega-dose treatment of vitamin C significantly attenuated FFD-mediated NAFLD by decreasing diet ingestion and increasing water intake. These data also suggested that the mega-dose vitamin C treatment facilitated the excretion of FFD-mediated purine derivatives in mice. Taken together, our study showed the beneficial effects of mega-dose vitamin C treatment in NAFLD, and we anticipated that the results from the present study would lead to a better understanding of the relationship between mega-dose vitamin C treatment and NAFLD.
Reference
- ASGHARPOUR, Amon, et al. A diet-induced animal model of non-alcoholic fatty liver disease and hepatocellular cancer. Journal of hepatology, 2016, 65.3: 579-588.
----------------------------------------------------------------------------------------
8. Please also indicate the limitations of this study.
Answer: Thanks to the reviewer’s helpful comments, we added limitations part in the discussion
(Newly added sentences, line 488-498): In the present study, we thoroughly investigate the effects of mega-dose vitamin C treatment in NAFLD progression. However, there were some limitations of our current experiments. First, we suggested that the mega-dose vitamin C treatment accelerated degradations of the FFD-mediated purine derivatives in mice. However, the precise mechanisms of mega-dose vitamin C treatment in uricase activities are still unknown. Seconds, all results observed in the present study come from animal experiments, and thus it can’t be directly applied to human beings because of the species differences. Given that, it seems that furthered studies are needed to confirm and expand these findings.
----------------------------------------------------------------------------------------
Conclusions
1. Please complete the conclusions.
Answer: Thanks to the reviewer’s helpful comments, we newly added the conclusions in the revised version of manuscript
(Newly added sentences, line 500-508): In summary, we have demonstrated that the mega-dose treatment of vitamin C significantly attenuated FFD-mediated NAFLD by decreasing diet ingestion and increasing water intake. These data also suggested that the mega-dose vitamin C treatment facilitated the excretion of FFD-mediated purine derivatives in mice. Taken together, our study showed the beneficial effects of mega-dose vitamin C treatment in NAFLD, and we anticipated that the results from the present study would lead to a better understanding of the relationship between mega-dose vitamin C treatment and NAFLD.
----------------------------------------------------------------------------------------
References
1. The way of citation needs to be improved
Answer: Thanks to the reviewer’s helpful comments, we revised the citation format using used EndNote manager software (EndNote X9 version)
The citation format was acquired from; https://endnote.com/style_download/nutrients/

Round 2
Reviewer 1 Report
The authors provided feedback to all concerns raised. The manuscript has been improved.
Reviewer 2 Report
Thanks to the authors for their revisions.